# Emotions and Motivations Underlying Adherence to the Anti-COVID-19 Vaccination Campaign: A Survey on a Sample of Italians under 30 Years

**DOI:** 10.3390/ijerph19010077

**Published:** 2021-12-22

**Authors:** Luna Carpinelli, Francesco De Caro, Giulia Savarese, Mario Capunzo, Monica Mollo, Giuseppina Moccia

**Affiliations:** 1Department of Medicine, Surgery and Dentistry—Scuola Medica Salernitana, University of Salerno, 84081 Baronissi, Italy; lcarpinelli@unisa.it (L.C.); fdecaro@unisa.it (F.D.C.); mcapunzo@unisa.it (M.C.); gmoccia@unisa.it (G.M.); 2Department of Human Science, University of Salerno, 84084 Fisciano, Italy; mmollo@unisa.it

**Keywords:** COVID-19 vaccine, adherence, young adult

## Abstract

Background: In Italy, the under-30 age category was the one that joined the anti-COVID-19 vaccination campaign in an important way. This study investigates the emotional states and motivations underlying joining the anti-COVID-19 vaccination campaign. Methods: A questionnaire consisting of SF-12, STAI Y, and open questions was administered to investigate the state of health, the state of anxiety, and motivational states of the participants. Results: Of the sample, 80.7% were vaccinated at the first call, deeming the action important to combat the infection. However, 48.2% stated that they were quite worried about the problems related to the pandemic, 37.3% feared being directly infected, and 43.4% were worried about the health of relatives and friends. Conclusions: The positive impact that the vaccination campaign has had on the under-30 category is very significant for the immunization process, which is of fundamental importance for fighting the pandemic, so the “benefits” outweigh the “risks” related to the COVID-19 vaccine.

## 1. Introduction

Recent studies [1,2,3,4,5] have found a greater availability and adherence to COVID-19 vaccination campaigns among the young (under-30), identifying, among the influencing factors, concerns of getting infected and transmitting the disease to family members and a high confidence in medical science and the actions of politicians in their country [6,7,8,9]. From an analysis of a data report [10], it emerged that in Italy, young people between 20 and 29 years old (for whom the vaccination campaign began in June 2021) were the most available category for vaccination, reaching an adherence percentage of 74% (first dose) and 65% (second dose) in just three months, compared to a percentage of 73% vaccinated for those over 30 years old who, on the contrary, received the call six months prior to them.

As is known, there have been several movements in favor of and against COVID-19 vaccination campaigns, which have based their ideas and positions on the basis of a right to health and decision-making freedom. All this has weakened the extensive vaccination campaign that has been conducted worldwide and which has highlighted the effectiveness of the vaccines against the spread of infection through scientific studies. Gallè et al. [11] investigated the knowledge and acceptance underlying the adherence to COVID-19 vaccination among Italian university students. The results of their work highlighted a high level of acceptance of COVID-19 vaccination and a good level of knowledge of the risks and benefits of the vaccines in this population group and that these variables correlate with each other. Despite the recent Italian studies conducted on the young adult population [6,11], there is still no evidence of the emotions and anxiety underlying the vaccination campaign. For this reason, our study aimed to investigate these aspects that are missing in the literature and that we believe are useful for the purposes of a much more impressive vaccination campaign.

Therefore, the purpose of this survey study conducted on a group of young adults (subjects under 30 years of age) who joined the anti-COVID-19 vaccination campaign was to evaluate (a) the perception of the state of health in relation to moods and emotions linked to the pandemic; (b) the level of the state of anxiety, worry, tension, fear, and indecision related to joining the vaccination campaign; (c) the reasons behind the decision to join the vaccination campaign.

## 2. Methods

### 2.1. Procedure

The survey was conducted at the COVID Vaccination Center located at the Educational Center for Health Professions of the University of Salerno within AOU “San Giovanni di Dio and Ruggi d’Aragona” of Salerno (Campania, Italy) in the reference period of June–August 2021.

The participants joined the vaccination campaign by registering through the national government platform promoted by the Ministry of Health. The Italian population between 19 and 30 years old is 7,261,822 [12]. The population of the Campania Region of the same age is 817,689 and of these, as of 8 December 2021, 84.4% are vaccinated [13]. Our survey sample chosen at random represents 10% of the Campania population of young adults. After the administration of the vaccine, the observation phase in the post-vaccination room was scheduled. During the expected waiting time according to the vaccination protocols (from 15 to 30 min depending on the risk of adverse reactions due to diseases or allergies), users were shown a QR code linked to a questionnaire on the Google Forms platform or, alternatively, hard copy was provided.

### 2.2. Instruments

The online questionnaire was created ad hoc according to the CHERRIES statement [14] and divided into three sections: (1) the purpose of the survey, informed consent, and authorization to process personal data, made anonymous for research purposes; (2) socio-personal data and clinical history and standardized scales, such as the Short Form Health Survey (SF-12) [15], for the assessment of one’s physical and mental states; the State–Trait Anxiety Inventory (STAI-Y) [16,17] to detect the level of the state of anxiety, understood as a feeling of insecurity and helplessness in the face of perceived damage that can lead to worry or to flight and avoidance; (3) items 21–28 were created ad hoc with the aim of better investigating the experience lived during the COVID-19 pandemic and the factors connected to it, such as concern for one’s own and others’ state of health (personal, family, and friends), use of devices and implementation of anti-contagion measures, and adherence to vaccination campaigns and underlying reasons. Completion of the questionnaire took approximately 10–15 min.

The items 21–28 was based on the available data regarding COVID-19 vaccination issues and vaccination hesitancy [18], as well as statements issued by national institutions [19]. Items in this section were drafted by a panel of experts comprised of one epidemiologist, one sociologist, one expert in vaccinology, and two psychologists.

### 2.3. Sample

Eighty-three users participated in this survey (F = 38; mean age = 22.23; SD = 4.3), 94% of whom received the first dose of Pfizer-BioNTech, 6%—a single administration. Of these, 65.4% belonged to the ministerial category with an age between 18 and 29 years, while 34.6% belonged to the category between 30 and 31 years of age. Regarding the level of education, 83.1% of the subjects had high school diploma, while 16.9% had a degree. A total of 51.8% were university students, 22.9% were employed by a public or private company, 14.5% were self-employed, and 10.8% were unemployed.

To determine the socioeconomic status of the participants, we reread the data in an aggregate form from the Strategic Orientation Document of the Urban Authority of Salerno [20] which shows a medium–high level present in the urban and provincial areas of Salerno, which refers health competence to the COVID Vaccination Center where the survey was conducted.

Finally, 4.9% of the total sample reported suffering from chronic diseases, and 4.9% complained of articular pain.

## 3. Results

### 3.1. Quantitative Analysis 

IBM SPSS v.23 software (IBM® SPSS® Italy) was used for the descriptive analysis of the variables investigated and the comparison between the means of the scores obtained from the administered tests. Of the sample (see Table 1), 43.4% reported a very good level of health (item No. 1 of SF-12). Specifically, for the category of 18–29 years, 24.5% answered “Excellent” against 14.3% of the over 30 years of age group. In the analysis of item No. 9 “I feel calm and peaceful,” 26.4% of the group of 18–29 years old responded “Some of the time,” while 35.7% of the over 30 years of age group declared “Most of the time.” Item No. 11 “I feel downhearted and blue” highlighted as the most representative answer for both groups “A little of the time.” With regard to the STAI-Y scale, the 18–29 years of age group turned out to be emotionally more sensitive to the state of anxiety (STAI-Y mean score = 40.42, SD = 9.7; over-30 = 37.71, SD = 8.7). In fact, 9.4% reported feeling “Not at all” secure (over 30 = 3.6%), 43.4% felt tense “Most of the time” (over 30 = 39.3%), 1.9% felt “Very much so” frightened and nervous (5.7%), and both groups reported much indecision at the same level.

Of the sample, 48.2% reported being quite worried about the problems related to the pandemic, 37.3%—to be personally affected by the pandemic for the next six months, 43.4%—quite worried that for the next six months the pandemic would also directly affect their relatives and friends.

The correlation coefficients are all significant (all *p*-values are <0.001) and all exceed 0.30; it can, therefore, be said that there appears to be one quite strong correlation between all items, which should investigate different aspects of the same construct.

### 3.2. Qualitative Analysis 

The T-Lab software was used for the analysis of the headwords of the answers to the open questions. From the co-word analysis (see Figure 1), it emerged that there are factors linked to emotional and social aspects (Y-axis), such as protection and feeling safe from the disease (both for oneself and for one’s family). The term “call” refers to the reservation for their age group (highlighted in green), indicative of waiting to be summoned for vaccination. In addition, along the X-axis are other factors related to cognitive and moral aspects, with vaccination considered a right (blue) and important (red) action.

## 4. Discussion

The study of Gallè et al. [10] offers a picture of vaccine acceptance and knowledge in a large sample of Italian undergraduates during the first phase of the COVID-19 immunization campaign. It confirms that knowledge and acceptance are strictly related, underlining the role of correct information in fighting vaccine hesitancy. Our study investigated emotions and anxiety correlated to COVID-19 vaccination.

In our previous studies [18,21,22], we highlighted the emotions and moods related to joining a vaccination campaign, noting multiple shades of negative emotions, such as fear, indecision, and anxiety, present among the ministerial categories who had already undergone administration during the first vaccination phase. From this study, it is evident that “benefits” outweigh the “risks” related to the COVID-19 vaccines for young people under 30 years of age, naturally predisposed to proactivity and resilience, overcoming so-called “emotional reasoning”.

The positive impact that the vaccination campaign has had on the under-30 age category is very significant for the immunization process, which is of fundamental importance in fighting the pandemic. In fact, 80.7% of the sample presented themselves at the first call received through the ministerial platform for vaccination, deeming the vaccine to be the most important action to be taken to combat infection (81%). Despite the high indices of the state of anxiety, tension, nervousness, and indecision, the desire to definitively resume one’s habits and therefore return to one’s life characterized by sociality and study and/or work exceeded the perception of vaccine-related risk.

As the issue of vaccines takes hold, in light of the results of trials, the issue of membership begins to be treated as “the other side of the coin” in the vaccination issue. Already, since the summer of 2020, scientific interest has deviated from a more general theme of experimentation to ponder the question of individual behavior and the question itself about it. In fact, there is no need to explain how vaccination attitudes have been flaunted in a highly polarized way in interactions on major social networks and beyond, even with the competition by fake news, pushing research to address the issue in a specific way. At this point, it should be stated that the references to vaccination hesitancy indicated above are not the result of a systematic investigation; rather, they are proposed to formulate a scenario with which the planning of vaccination campaigns for COVID-19 will have to be addressed, or the “resistance behavior” of people.

In previous studies, we were interested in the emotional and cognitive implications underlying adherence to the vaccine and the limitation and inspiration for future studies is to be able to interview subjects who have not adhered to the vaccination campaign and compare the emotions and beliefs of this group.

## 5. Conclusions

Our study found high acceptance of COVID-19 vaccination among Italian young adults. This supports the effectiveness of the information strategy of the COVID-19 immunization campaign in Italy. Policymakers and communications of government officials and the media should pay attention to the spread of data not supported by scientific evidence. Therefore, on the basis of the results that emerged from our work, it is clear that the use of strategic and effective communications must be followed and treated on the basis of micro-communities [23,24] that are created in certain situations, which not only inform about vaccinations (risks and benefits), but also respond to doubts and fears of people who, submerged by a myriad of different sources, are unable to form their own opinion that can allow them to trust science and increasingly rapid developments in research [11,18].

## Figures and Tables

**Figure 1 ijerph-19-00077-f001:**
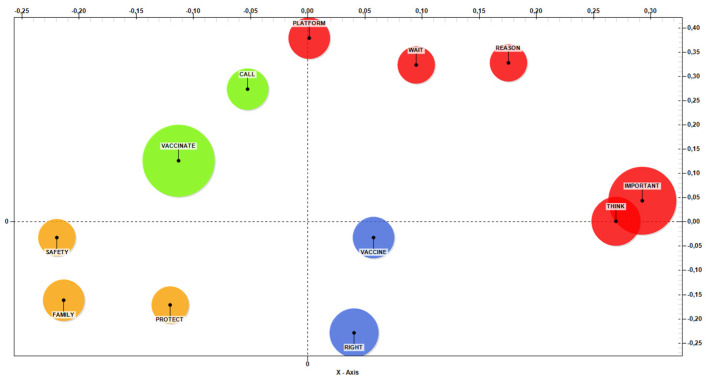
Co-word analysis (MDS = Sammon method; stress = 0.0247).

**Table 1 ijerph-19-00077-t001:** Percentage of the frequencies of responses to specific items of the questionnaire.

SF-12—Item No. 9, Calm and Peaceful?							
	All of the Time	Most of the Time	A good Bit of the Time	Some of the Time	A Little of the Time	None of the Time	
**Group aged** **18–29 years**	9.4%	30.2%	20.8%	26.4%	11.3%	1.9%	
**Group aged over 30 years**	21.4%	35.7%	17.9%	14.3%	10.7%		
**SF-12—Item No. 11, Downhearted and blue?**							
**Group aged** **18–29 years**		11.5%	9.6%	28.8%	42.3%	7.7%	
**Group aged over 30 years**				32.1%	42.9%	25%	
**STAI-Y—Item No. 2, I feel secure**							
	**Not at all**	**Somewhat**	**Moderately so**	**Very much so**			
**Group aged** **18–29 years**	9.4%	11.3%	64.2%	15.1%			
**Group aged over 30 years**	3.6%	14.3%	67.9%	14.3%			
**STAI-Y—Item No. 3, I feel tense**							
**Group aged** **18–29 years**	30.2%	43.4%	24.5%	1.9%			
**Group aged over 30 years**	32.1%	39.3%	10.7%	17.9%			
**STAI-Y—Item No. 9, I feel frightened**							
**Group aged** **18–29 years**	56.6%	32.1%	9.4%	1.9%			
**Group aged over 30 years**	82.1%	17.9%					
**STAI-Y—Item No. 12, I feel nervous**							
**Group aged** **18–29 years**	28.3%	49.1%	17%	5.7%			
**Group aged over 30 years**	46.4%	42.9%	10.7%				
**STAI-Y —Item No. 14, I feel indecisive**							
**Group aged** **18–29 years**	48.2%	23.1%	26.9%	3.8%			
**Group aged over 30 years**	64.3%	25%	7.1%	3.6%			
**Item 21. How concerned are you personally about the problems related to the COVID-19 pandemic at the moment?**							
	**Not at all worried**	**A little worried**	**Neither very nor a little worried**	**Quite worried**	**Very worried**	**Definitely worried**	**Extremely worried**
**Total group**	6%	4.8%	19.3%	48.2%	12%	8.4%	1.2%
**Item 22. How likely do you think you will be directly and personally affected by the COVID-19 pandemic in the next six months?**							
**Total group**	4.8%	13.3%	24.1%	37.3%	13.3%	6%	1.2%
**Item 23. How likely are your friends and family in the country where you currently live to be directly affected by the COVID-19 pandemic in the next six months?**							
**Total group**	3.6%	15.7%	20.5%	43.4%	13.3%	3.6	

## Data Availability

Written informed consent was obtained from the subjects in order to publish this paper.

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
