# Peer review of "Emotions and Motivations Underlying Adherence to the Anti-COVID-19 Vaccination Campaign: A Survey on a Sample of Italians under 30 Years"

_ijerph, 2021, doi:10.3390/ijerph19010077_

Round 1
Reviewer 1 Report
This paper studies the motions and motivations underlying adherence to the anti-COVID-19 vaccination campaign of the people aged under 30 years in Italy. The topic is interesting an of great importance under the COVID-19 epidemic. However, there are also some shortcomings that need to be modified as following. (1) How to take effective measures to improve the vaccination willingness based on the results? More policy implications can be disscused in the conclusion part. (2) Through the current description, we cannot see the representativeness and typicality of the sample. More detailed information of the sample , such as the gender, education level, economic situation and geographical distribution, should be described in the sample part. (3) Compared with the people under the age of 30, the coverage of the people over 30 years olds are too wide (including middle-aged and aged persons). A direct comparison between the two two groups of people may not be appropriate.Author Response
see text in yellow colorWe thank the reviewer for the positive feedback on our work and for the valuable suggestions we followed in trying to improve the paper. We answer in detail:
1) in the discussion and conclusions paragraph we have included a useful strategy to effectively improve the vaccination campaign.
2) In the descriptive paragraph the characteristics of the sample, we have inserted the variables of education, work employment and specified the territorial socio-economic status (SES) of the participants belonging to the vaccination center as they are of local health competence.
3) We thank the referee for highlighting this detail. In reality, the group of subjects over 30s includes only participants aged 30-31 years. Furthermore, for the specification of the groups we used the ministerial categories of the Italian vaccination campaign (https://www.salute.gov.it/portale/nuovocoronavirus/dettaglioContenutiNuovoCoronavirus.jsp?lingua=italiano&id=5452&area=nuovoCoronavirus&menu=vuoto).
We have best specified this information in the "Sample" paragraph.
Reviewer 2 Report
„The survey was conducted in the reference period June–August 2021 at the COVID 36 Vaccination Center at the A.O.U. "San Giovanni di Dio and Ruggi d’Aragona" of Salerno (Campania, Italy). Following inoculation of the vaccine and during the expected waiting time according to the vaccination protocol (from 15 to 30 min depending on the risk of adverse reactions due to diseases or allergies), users were shown a QR code linked to the questionnaire on the Google Forms platform or, alternatively, were provided a copy in paper form.“ The study included only individuals that had gotten their vaccine. There is an obvious sampling bias. No data are available for the vaccine hesitant group, thus a comparison is not possible. I am not certain what the aim of the study is.
Author Response
see text in green colorWe thank the reviewer for the correct reflection to which we have responded by detailing the objectives of our study in the introduction pragraph and setting the possible sampling of subjects not adhering to the vaccination campaign as a limit and a future starting point.
In previous studies we have already been interested in the emotional and cognitive implications underlying adherence to the vaccine.
Introduction: […] As is known, there have been several movements in favor of and against the COVID-19 vaccination campaign, which have based their ideas and positions on the basis of the right to health and the right of decision-making freedom. All this has weakened the extensive vaccination campaign that has been conducted worldwide and which has highlighted the effectiveness of the vaccine against the spread of the infection through scientific studies.
The objectives of this survey study were the evaluation in the group of subjects under 30 years adhering to the anti COVID-19 vaccination campaign of (a) the perception of the state of health also in relation to the moods and emotions related to the pandemic; (b) the level of state anxiety, worry, tension, fear, and indecision connected to joining the vaccination campaign; (c) the reasons underlying the decision to join the vaccination campaign.
Discussion and copnclusions: […] A limit and inspiration for future studies is to be able to interview subjects who have not adhered to the vaccination campaign and compare the emotions and beliefs of the group. of the vaccinated.
Round 2
Reviewer 1 Report
I think this paper is significantly improved after the revision and can be accepted in present form.
Author Response
Dear reviewer, thank you!Reviewer 2 Report
The issues were not adequately addressed.
Author Response
Dear review, thanks for your suggestions. We have tried to respond to all critical issues.
